# MaxSAT-Based Compression for Tsetlin Machines

**Stefan Szeider** [1]

## Abstract

We consider the computational problem of compacting Tsetlin Machine classifiers by reducing the number of propositional clauses while preserving predictive accuracy. TMs trained with limited clause capacity often perform poorly because stochastic optimization cannot reliably find the few precise clauses needed in a vast configuration space. High-quality compact subsets also exist in the case of larger Tsetlin Machines. The difficulty here lies in extracting them. Local pruning heuristics can fail badly on TMs because clauses interact through Boolean logic: a clause may appear unimportant in isolation yet becomes critical when others are removed. We formalize compression as the Minimum Discriminating Clause Set (MDCS) problem, which asks to find a smallest subset of clauses that preserves the trained model's discrimination of training samples. We show that MDCS is NP-hard. We solve MDCS using weighted partial Maximum Satisfiability (MaxSAT). A partition-and-merge strategy allows us to scale to 100,000 samples. Across 13 datasets, the compressed model preserves the 200-clause teacher's accuracy within a few percentage points while using a median of only 16 clauses, and outperforms a matched-capacity TM trained from scratch on every dataset where direct training has room to improve, by up to 45 percentage points.

## 1. Introduction

Tsetlin Machines (TMs) are an alternative to neural networks for pattern classification (Granmo, 2018). Where neural networks learn continuous weight matrices, TMs learn decision rules that can be expressed as propositional logic clauses. Each clause consists of a conjunction of literals, such as $(x_3 \land \neg x_7 \land x_{12})$, that fires when its conditions match an input. The final prediction is obtained by aggregating votes from clauses assigned to each class. This architecture has two properties that make it suitable for applications where neural networks fall short: the learned rules are propositional clauses that can be inspected directly, and inference requires only bitwise operations. The TM architecture is particularly suited for domains that require transparency (e.g., medical diagnosis, regulatory compliance) and for deployment on resource-constrained hardware (e.g., FPGAs, microcontrollers) (Wheeldon et al., 2020; Duan et al., 2025).

Roughly speaking, a TM clause is built from teams of Tsetlin Automata (TAs), finite state machines that are connected by stochastic reward/penalty feedback. One distinguishes between Type I and Type II feedback; Type I reinforces literals that capture target-class patterns, while Type II suppresses clauses that fire on counterexamples. This learning mechanism provably converges to Nash equilibria under certain conditions (Granmo, 2018). In experiments with tabular data, TMs are competitive with state-of-the-art gradient-based methods on both text classification and image tasks when combined with a convolutional structure (Granmo et al., 2019; Abeyrathna et al., 2021).

**The compact-model problem.** Training compact models is harder than training large ones if it requires avoiding suboptimal solutions that stochastic optimization tolerates when capacity is abundant. This tension is particularly pronounced for TMs. On the spect-heart dataset, a TM trained with 6 clauses achieves only 39.1% accuracy, worse than random guessing.

Why does training small fail? We hypothesize that with few clauses, each must land precisely in a vast configuration space, a haystack; the stochastic search may rarely find the needle. This parallels the lottery ticket phenomenon in neural networks (Frankle & Carbin, 2019): winning substructures exist within over-parameterized models, but finding them requires training large models first. The same holds for TMs, because the combinatorial structure of propositional logic enables *exact* extraction via MaxSAT rather than magnitude-based heuristics.

**Our approach.** We sidestep the small-model training problem. We train a TM with excess capacity. We then find a smallest clause subset that preserves the trained model's

---

[1]Algorithms and Complexity Group, TU Wien, Vienna, Austria. Correspondence to: Stefan Szeider <sz@ac.tuwien.ac.at>.

*Proceedings of the 43rd International Conference on Machine Learning*, Seoul, South Korea. PMLR 306, 2026. Copyright 2026 by the author(s).

discrimination of samples. For each pair of training samples the model classifies differently, at least one selected clause must output differently on them. We will refer to this as the *Minimum Discriminating Clause Set* (MDCS) problem. We show that MDCS is NP-hard. We encode it as a weighted partial MaxSAT problem, where hard constraints enforce discrimination and soft constraints minimize the number of selected clauses.

The number of constraints defined this way is $O(n^2)$, one per sample pair, which overwhelms solvers beyond a few thousand samples. Following the incremental partitioning approach of Ghosh & Meel (2019), we partition the training set into groups and solve MaxSAT independently for each group, merging the selected clauses to scale exact optimization to large datasets. Partitioning reduces constraints by a factor equal to the number of partitions. Separation is guaranteed within each partition, and the union of solutions empirically preserves cross-partition separations. This approach scales to datasets of 100K samples in approximately two minutes, a regime where naive encoding is intractable.

A 200-clause TM on banknote compresses to 4 clauses with the same 86.3% accuracy. On spect-heart it compresses to 6 clauses and reaches 84.4%, above the 200-clause teacher's 83.7%. A TM trained from scratch at the same budgets achieves only 71.1% and 39.1%. On breast-cancer, compression from 200 to 12 clauses gives 91.4% accuracy compared to 63.5% for a 12-clause TM trained from scratch. These are not outliers: across 13 datasets, the compressed model matches the 200-clause teacher within a few percentage points while substantially outperforming the matched-capacity baseline on every dataset where direct training has room to improve. To the best of our knowledge, our work is the first formal optimization approach to TM compression.

An alternative approach would be to prune a large trained TM. Unfortunately, greedy pruning (iteratively removing the "least important" clause) fails badly on some datasets. On banknote, greedy pruning collapses to 64.4%, 22pp below MaxSAT. The gap is consistent with how TM clauses interact through Boolean logic and vote aggregation: a clause that appears unimportant in isolation may become critical once others are removed. A similar phenomenon was previously observed by CHITA (Benbaki et al., 2023) in the context of neural networks. The main difference is that TM clauses interact through Boolean logic, making the combinatorial structure explicit and exactly solvable via constraint satisfaction.

## 2. Background

### 2.1. Tsetlin Machines

Let $V$ be a finite set of *propositional variables*. An *input* is a vector $\mathbf{x} \in \{0,1\}^V$. A *literal* over $V$ is either a variable $v \in V$ or its negation $\neg v$. A *clause* $c$ is a set of literals; it *evaluates to 1* under $\mathbf{x}$, written $c(\mathbf{x}) = 1$, if every literal in $c$ is satisfied by $\mathbf{x}$.

A *Tsetlin Machine* (TM) for binary classification is a pair $T = (\mathcal{C}^+, \mathcal{C}^-)$ of disjoint finite clause sets (Granmo, 2018). We write $\mathrm{var}(T)$ for the set of propositional variables appearing in the clauses of $T$. For an input $\mathbf{x} \in \{0,1\}^{\mathrm{var}(T)}$, the TM predicts

$$T(\mathbf{x}) = \mathrm{sign}\left( \sum_{c \in \mathcal{C}^+} c(\mathbf{x}) - \sum_{c \in \mathcal{C}^-} c(\mathbf{x}) \right) \in \{-1, +1\}, \tag{1}$$

where $v$ denotes the vote sum and $\mathrm{sign}(v) = +1$ if $v \geq 0$, and $-1$ otherwise. This corresponds to the standard TM output convention $u(\mathbf{x}) \in \{0,1\}$, where $u(\mathbf{x}) = 1$ iff the vote sum is nonnegative, under the encoding $y \mapsto 2y - 1$ from $\{0,1\}$ to $\{-1,+1\}$.

Clause construction is specified by teams of Tsetlin Automata (TAs), one TA for each clause-literal pair, deciding whether to *include* or *exclude* that literal. Type I feedback reinforces literals that capture target-class patterns, while Type II feedback suppresses clauses that evaluate to 1 on counterexamples. TM clauses are human-readable rules, but they interact nonlinearly through Boolean logic and vote aggregation, so the predictive effect of a clause is inherently context-dependent.

### 2.2. Weighted MaxSAT

A *weighted partial MaxSAT instance* consists of a set $H$ of *hard* clauses, a set $S$ of *soft* clauses, and a weight function $w$ assigning a positive weight to each soft clause. A *solution* is a truth assignment $\tau$ that satisfies all hard clauses. The *cost* of a solution is the total weight of soft clauses violated:

$$\mathrm{cost}(\tau) = \sum_{\omega \in S,\, \tau \not\models \omega} w(\omega). \tag{2}$$

An *optimal* solution minimizes cost, or equivalently, maximizes the total weight of satisfied soft clauses. Our approach is based on RC2 (Ignatiev et al., 2019), an efficient "core-guided" MaxSAT solver that terminates with a globally optimal solution.

### 2.3. Related Work

**TM Compression.** Previous work on TM compression uses heuristic training modifications or local pruning. The Weighted TM assigns real-valued clause weights so that strong clauses replace many redundant unit-vote clauses (Phoulady et al., 2019); a variant uses integer weights for improved interpretability (Abeyrathna et al., 2021). ETHEREAL prunes literal and automata states and applies compact encodings for TinyML deployment (Duan

et al., 2025). Knowledge distillation transfers behavior from large teacher TMs to smaller students via soft-label generation (Kinateder, 2025). These methods can reduce model size. To the best of our knowledge, there has been no prior work posing clause selection as a *global* formal optimization problem with exact feasibility constraints.

**Neural Network Pruning.** Magnitude-based pruning removes small weights (Han et al., 2015), but TMs lack analogous continuous parameters. Recent combinatorial pruning work shows that component importance depends on context: components unimportant in isolation can become important after other components are removed. CHITA (Benbaki et al., 2023) uses combinatorial optimization to frame this problem explicitly, a conceptually related approach.

**Rule Learning via MaxSAT.** Ghosh & Meel (2019) use weighted MaxSAT to *learn new* sparse CNF classifiers from data. In contrast, we *select* a minimal subset from an already-trained TM while preserving its decision behavior. Our approach differs in constraint structure (selection over a fixed clause pool) and in interpretation (model compression vs. de novo rule induction). However, we closely follow their incremental partitioning approach for scalability.

## 3. Problem Formulation

### 3.1. Oracle-Based Compression

Let $T = (\mathcal{C}^+, \mathcal{C}^-)$ be a trained Tsetlin Machine, and let $\mathcal{X} \subseteq \{0,1\}^{\mathrm{var}(T)}$ be the set of training samples. We treat $T$ as an *oracle*. The goal is now to find a smallest subset $\mathcal{K} \subseteq \mathcal{C}^+ \cup \mathcal{C}^-$ for which the *compressed TM* $T_\mathcal{K} = (\mathcal{K} \cap \mathcal{C}^+, \mathcal{K} \cap \mathcal{C}^-)$ preserves $T$'s predictions on $\mathcal{X}$.

We compress the learned decision behavior encoded by $T$ and do *not* re-fit the original labels. This oracle-based view offers an advantage: training benefits from redundancy, while compression selects a small subset that preserves the learned separations.

### 3.2. Minimum Discriminating Clause Set (MDCS)

Let $T = (\mathcal{C}^+, \mathcal{C}^-)$ be a TM. For any two samples $\mathbf{x}, \mathbf{x}' \in \{0,1\}^{\mathrm{var}(T)}$, we define the set of *separating clauses*

$$\mathrm{Sep}(\mathbf{x}, \mathbf{x}') = \{\, c \in \mathcal{C}^+ \cup \mathcal{C}^- \mid c(\mathbf{x}) \neq c(\mathbf{x}') \,\}.$$

For any set of samples $S \subseteq \{0,1\}^{\mathrm{var}(T)}$, we define the set of *discriminating pairs*

$$\mathcal{P}(S) = \{\, \{\mathbf{x}, \mathbf{x}'\} \mid \mathbf{x}, \mathbf{x}' \in S, \ T(\mathbf{x}) \neq T(\mathbf{x}') \,\}.$$

**Definition 3.1** (MDCS). Given a TM $T = (\mathcal{C}^+, \mathcal{C}^-)$ and a set of samples $\mathcal{X} \subseteq \{0,1\}^{\mathrm{var}(T)}$, the *Minimum Discriminating Clause Set* problem asks for a smallest

$\mathcal{K} \subseteq \mathcal{C}^+ \cup \mathcal{C}^-$ such that for every $\{\mathbf{x}, \mathbf{x}'\} \in \mathcal{P}(\mathcal{X})$ we have $\mathcal{K} \cap \mathrm{Sep}(\mathbf{x}, \mathbf{x}') \neq \emptyset$.

Intuitively, selected clauses can preserve $T$'s behavior as long as at least one clause in $\mathcal{K}$ outputs differently on $\mathbf{x}$ and $\mathbf{x}'$ for every pair that $T$ *separates*.

**Proposition 3.2.** *Let $\mathcal{K}$ be a feasible solution to MDCS. For any $\mathbf{x}, \mathbf{x}' \in \mathcal{X}$ with $T(\mathbf{x}) \neq T(\mathbf{x}')$, there exists $c \in \mathcal{K}$ with $c(\mathbf{x}) \neq c(\mathbf{x}')$.*

*Proof.* Since $\{\mathbf{x}, \mathbf{x}'\} \in \mathcal{P}(\mathcal{X})$, feasibility requires $\mathcal{K} \cap \mathrm{Sep}(\mathbf{x}, \mathbf{x}') \neq \emptyset$. Any $c$ in this intersection satisfies $c(\mathbf{x}) \neq c(\mathbf{x}')$ by definition of Sep. $\square$

*Remark* 3.3. For any $\{\mathbf{x}, \mathbf{x}'\} \in \mathcal{P}(\mathcal{X})$, the set $\mathrm{Sep}(\mathbf{x}, \mathbf{x}')$ is nonempty: since $T(\mathbf{x}) \neq T(\mathbf{x}')$, the clause output sums must differ, so at least one clause outputs differently on $\mathbf{x}$ and $\mathbf{x}'$. Thus, the hard constraints in the MaxSAT encoding are never empty disjunctions.

**Theorem 3.4.** *MDCS is NP-hard.*

*Proof.* We reduce from MINIMUM HITTING SET, which asks for a smallest subset $H$ of a finite universe $U$ that intersects every member of a given collection $\mathcal{S}$ of nonempty subsets of $U$.

Given an instance $(U, \mathcal{S})$ with $|U| = n$, we construct an MDCS instance as follows. For each element $u \in U$, create a single-literal clause $c_u$ and place it in $\mathcal{C}^+$. For each set $S \in \mathcal{S}$, create a set $E_S$ of $n$ fresh single-literal clauses and place them in $\mathcal{C}^-$, ensuring all $E_S$ are pairwise disjoint. The sample set is $\mathcal{X} = \{a\} \cup \{b_S : S \in \mathcal{S}\}$, where sample $a$ fires every clause in $\mathcal{C}^+$ and no clause in $\mathcal{C}^-$, while the sample $b_S$ fires $c_u$ iff $u \notin S$, fires every clause in $E_S$, and fires no clause in $E_{S'}$ for $S' \neq S$.

On sample $a$ the vote is $n - 0 > 0$, so $T(a) = +1$. On sample $b_S$ the vote is $(n - |S|) - n = -|S| < 0$ (using $S \neq \emptyset$), so $T(b_S) = -1$. Since all samples $b_S$ share the same prediction, the discriminating pairs are precisely $\mathcal{P}(\mathcal{X}) = \{\{a, b_S\} : S \in \mathcal{S}\}$. For such a pair, clause $c_u$ fires on $a$ but not on $b_S$ exactly when $u \in S$, and clauses in $E_S$ fire on $b_S$ but not on $a$, giving $\mathrm{Sep}(a, b_S) = \{c_u : u \in S\} \cup E_S$.

Any hitting set $H$ yields a feasible solution $\mathcal{K} = \{c_u : u \in H\}$ with $|\mathcal{K}| = |H|$. Conversely, given a feasible $\mathcal{K}$, the sets $E_S$ are pairwise disjoint, so each $S$ with $\mathcal{K} \cap E_S \neq \emptyset$ contributes at least one clause to $\mathcal{K}$. Replacing all such clauses with a single $c_u$ for some $u \in S$ preserves feasibility without increasing size: clauses in $E_S$ contribute only to $\mathrm{Sep}(a, b_S)$ (because $c$ does not fire on $b_{S'}$ for $S' \neq S$), so swapping $E_S$ members for $c_u$ leaves the discriminating set for every other pair untouched while still covering $\mathrm{Sep}(a, b_S)$, which contains $c_u$. After this replacement, $\mathcal{K}$ contains only clauses of the form $c_u$, and $H = \{u : c_u \in \mathcal{K}\}$ is a hitting set with $|H| \leq |\mathcal{K}|$. $\square$

## 3.3. MaxSAT Encoding

We can encode MDCS to Weighted Partial MaxSAT (allowing *hard* clauses that always need to be satisfied and *soft* clauses that are subject to optimization; soft clauses carry positive integer weights). For each clause $c \in \mathcal{C}^+ \cup \mathcal{C}^-$, we introduce the Boolean *selection variable* $s_c$, where $s_c = 1$ means $c \in \mathcal{K}$. For every discriminating pair $\{\mathbf{x}, \mathbf{x}'\} \in \mathcal{P}(\mathcal{X})$, we add the hard clause

$$\bigvee_{c \in \text{Sep}(\mathbf{x}, \mathbf{x}')} s_c, \qquad (3)$$

which ensures that at least one separating clause is selected. For each $c \in \mathcal{C}^+ \cup \mathcal{C}^-$, we add a soft clause $(\neg s_c, w_c)$ of weight $w_c > 0$ to favor small solutions (default: uniform $w_c \equiv 1$; see Section 4.2 for an optional refinement). With uniform weights, maximizing satisfied soft clauses is equivalent to minimizing $|\mathcal{K}|$ subject to (3).

The number of potential constraints is $O(|\mathcal{X}|^2)$, but only pairs in $\mathcal{P}(\mathcal{X})$ contribute. Many pairs yield the same separating set because samples often induce identical clause-output patterns; in practice, we eliminate repeated hard clauses before calling the solver.

# 4. MaxSAT Compression Algorithm

## 4.1. Basic MaxSAT Compression

**Patterns.** Let $T = (\mathcal{C}^+, \mathcal{C}^-)$ be a TM and $\mathcal{K} \subseteq \mathcal{C}^+ \cup \mathcal{C}^-$. For $\mathbf{x} \in \{0, 1\}^{\text{var}(T)}$, the *pattern* of $\mathbf{x}$ under $\mathcal{K}$ is the function $\pi(\mathbf{x}, \mathcal{K}) : \mathcal{K} \to \{0, 1\}$ with $\pi(\mathbf{x}, \mathcal{K})(c) = c(\mathbf{x})$. The pattern $\pi(\mathbf{x}, \mathcal{K})$ can be seen as a bit vector of length $|\mathcal{K}|$ recording which selected clauses evaluate to 1. We write $U = |\{\pi(\mathbf{x}, \mathcal{K}) \mid \mathbf{x} \in \mathcal{X}\}|$ for the number of distinct training patterns.

**Prediction at test time.** Given the compressed model $(\mathcal{K}, D)$ returned by Algorithm 1, we classify a new sample $\mathbf{x} \in \{0, 1\}^{\text{var}(T)}$ as follows:

1. Compute $\pi(\mathbf{x}, \mathcal{K})$.

2. If this pattern is a key in $D$, return $D[\pi(\mathbf{x}, \mathcal{K})]$.

3. Otherwise, fall back to 1-nearest neighbor in Hamming space: return $D[\pi^\star]$ where

$$\pi^\star \in \arg\min_{\pi \in \text{keys}(D)} d_H(\pi(\mathbf{x}, \mathcal{K}), \pi).$$

Here $d_H$ denotes Hamming distance, with ties broken arbitrarily.

**Why not use $T_\mathcal{K}$ directly?** While the compressed TM $T_\mathcal{K} = (\mathcal{K} \cap \mathcal{C}^+, \mathcal{K} \cap \mathcal{C}^-)$ is a valid Tsetlin Machine, its

---

**Algorithm 1** MaxSAT-TM compression (training-time).

**Require:** Trained TM $T = (\mathcal{C}^+, \mathcal{C}^-)$; training samples $\mathcal{X} \subseteq \{0, 1\}^{\text{var}(T)}$
**Ensure:** Selected clauses $\mathcal{K} \subseteq \mathcal{C}^+ \cup \mathcal{C}^-$; pattern dictionary $D$
1: Compute predictions $T(\mathbf{x})$ for all $\mathbf{x} \in \mathcal{X}$
2: Compute discriminating pairs $\mathcal{P}(\mathcal{X})$ and separating sets $\text{Sep}(\mathbf{x}, \mathbf{x}')$
3: Create a Boolean variable $s_c$ for each $c \in \mathcal{C}^+ \cup \mathcal{C}^-$
4: Build MaxSAT instance:
5:    **Hard:** for each $\{\mathbf{x}, \mathbf{x}'\} \in \mathcal{P}(\mathcal{X})$, add $\bigvee_{c \in \text{Sep}(\mathbf{x}, \mathbf{x}')} s_c$
6:    **Soft:** for each $c \in \mathcal{C}^+ \cup \mathcal{C}^-$, add $(\neg s_c, w_c)$   // *default:* $w_c \equiv 1$
7: Solve MaxSAT; set $\mathcal{K} \leftarrow \{c \in \mathcal{C}^+ \cup \mathcal{C}^- \mid s_c = 1\}$
8: Build $D$: for each $\mathbf{x} \in \mathcal{X}$, store key $\pi(\mathbf{x}, \mathcal{K})$ with value $T(\mathbf{x})$
9: **return** $(\mathcal{K}, D)$

---

vote counts may differ from $T$'s even when patterns are preserved. For example, $T(\mathbf{x}) = +1$ might encode a vote of $5 - 3 = +2$, but under $\mathcal{K}$, the vote could become $2 - 3 = -1$. The dictionary $D$ contains the *original* predictions $T(\mathbf{x})$, so that the compressed model reproduces $T$ exactly on all training samples.

**Dictionary consistency.** The hard constraints (3) ensure that no two samples $\mathbf{x}, \mathbf{x}' \in \mathcal{X}$ with $T(\mathbf{x}) \neq T(\mathbf{x}')$ share the same pattern under $\mathcal{K}$ (see Proposition 3.2). This means that $D$ is well-defined. If $\pi(\mathbf{x}, \mathcal{K}) = \pi(\mathbf{x}', \mathcal{K})$, then $\{\mathbf{x}, \mathbf{x}'\} \notin \mathcal{P}(\mathcal{X})$, and so $T(\mathbf{x}) = T(\mathbf{x}')$.

**Why a hybrid predictor?** The MaxSAT constraints guarantee discrimination on *training* samples with different $T$-predictions, but not on *test* inputs that induce previously unobserved patterns. The 1-NN fallback provides predictions for unseen patterns by finding the closest training pattern in Hamming space.

## 4.2. Validation-Guided Weighting

Uniform weights target a smallest discriminating set. Not all clauses share the same generalization ability: some capture patterns that transfer to unseen data while others may memorize training-set idiosyncrasies. Optionally, we can bias the objective to retain empirically useful clauses.

**Weight computation.** The following procedure computes clause weights $w_c$ for the soft clauses in line 6 of Algorithm 1, replacing the default uniform weights.

Let $T = (\mathcal{C}^+, \mathcal{C}^-)$ be a TM and $\mathcal{X} \subseteq \{0, 1\}^{\text{var}(T)}$ a set of training samples.

1. Split $\mathcal{X}$ into $\mathcal{X}_{\text{train}}$ (80%) and $\mathcal{X}_{\text{val}}$ (20%).

2. Train $T$ on $\mathcal{X}_{\text{train}}$.

3. For each clause $c \in \mathcal{C}^+ \cup \mathcal{C}^-$, compute *discriminative utility*:

$$u(c) := |\{\, \{\mathbf{x}, \mathbf{x}'\} \in \mathcal{P}(\mathcal{X}_{\text{val}}) \mid c(\mathbf{x}) \neq c(\mathbf{x}') \,\}|.$$

4. Set $w_c = u(c) + 1$ so high-utility clauses are "expensive" to remove.

The main technical contribution of this work is the MaxSAT feasibility formulation; weighting is an *optional refinement*. In preliminary experiments, we found that weighting sometimes improved accuracy on datasets with high-dimensional features and limited samples.

### 4.3. Incremental Partitioning for Scale

A direct encoding uses one hard clause per discriminating pair, yielding $O(|\mathcal{X}|^2)$ constraints and becoming intractable beyond $|\mathcal{X}| \approx 5\text{K}$. To address this issue, we use partition-and-merge (Ghosh & Meel, 2019).

**Partition-and-merge.** When $\mathcal{X}$ is too large for direct MaxSAT encoding, we compute the clause subset $\mathcal{K}$ (line 7 of Algorithm 1) using the following procedure.

Let $T = (\mathcal{C}^+, \mathcal{C}^-)$ be a TM and $\mathcal{X} \subseteq \{0, 1\}^{\text{var}(T)}$ a set of training samples.

1. Randomly partition $\mathcal{X}$ into $p$ disjoint subsets, each of size $\sim |\mathcal{X}|/p$.

2. For each subset $\mathcal{X}'$ in the partition, solve the MDCS problem for $\mathcal{P}(\mathcal{X}')$, obtaining a clause subset $\mathcal{K}' \subseteq \mathcal{C}^+ \cup \mathcal{C}^-$.

3. Merge: set $\mathcal{K}$ to the union of all clause subsets from step (2).

4. Optionally refine iteratively until $\mathcal{K}$ stabilizes (not used in our experiments).

**Complexity.** Each partition contributes $O((|\mathcal{X}|/p)^2)$ constraints; the total number across $p$ partitions is $O(|\mathcal{X}|^2/p)$, a factor-$p$ reduction. The merged solution $\mathcal{K}$ guarantees separation within each partition; cross-partition separation is not formally guaranteed but holds empirically. With $p = 128$, we compress Higgs-100K ($|\mathcal{X}| = 100{,}000$) in approximately 2 minutes.

*Table 1.* Dataset characteristics. The top 10 are the main evaluation datasets; the bottom 3 are near-saturated.

| DATASET | SAMPLES | FEATURES | SOURCE |
|---|---|---|---|
| SPECT-HEART | 267 | 22 | UCI |
| BREAST-CANCER | 569 | 30 | UCI |
| TICTACTOE | 958 | 9 | UCI |
| BANKNOTE | 1,372 | 4 | UCI |
| KR-VS-KP | 3,196 | 36 | UCI |
| SPAMBASE | 4,601 | 57 | UCI |
| PHISHING | 11,055 | 10 | OPENML |
| MAGIC | 19,020 | 10 | UCI |
| ELECTRICITY | 45,312 | 8 | OPENML |
| HIGGS-100K | 100,000 | 28 | UCI |
| NURSERY | 12,960 | 8 | UCI |
| MUSHROOM | 8,124 | 22 | UCI |
| CAR | 1,728 | 6 | UCI |

## 5. Experiments

### 5.1. Experimental Setup

**Datasets.** We evaluate on 13 binary classification datasets from UCI and OpenML repositories (Table 1), selected to cover a range of sizes, feature counts, and domains. These datasets range in size from 267 samples (spect-heart) to 100,000 (higgs-100k), with feature counts from 4 (banknote) to 57 (spambase). Domains include medical diagnosis, fraud detection, game playing, and particle physics. Three datasets (nursery, mushroom, and car) are near-saturated, and both methods achieve over 90% accuracy. We obtain binary features from continuous features by thresholding at the median, and from categorical features by one-hot encoding.

**Tsetlin Machine Configuration.** All TMs use 200 clauses (100 per class), threshold $T = 5000$, specificity $s = 10.0$, the Weighted TM variant (Phoulady et al., 2019), and train for 100 epochs. Note that "weighted" here refers to the Weighted TM architecture (integer per-clause weights inside the TM voting rule); it is unrelated to the MaxSAT soft-clause weights $w_c$ used in Section 3.3. Tables 2–4 use validation-guided weights (Section 4.2); we report this convention here so that the figures are reproducible from the released scripts. We refer to this 200-clause configuration as the "full TM." For each dataset, $p$ ranges from 16 (small datasets) to 128 (higgs-100k), chosen to keep per-partition sample counts below 5,000 for solver tractability.

**Per-dataset exception: higgs-100k.** On higgs-100k, $T = 5000$ produces an over-specialized teacher whose discriminating-pair instance does not solve in any reasonable time (RC2 fails to finish a single partition within 30 min). For this dataset only, we train teacher TMs at $T = 15$, $s = 3.0$, which yields a comparable teacher accuracy ($\sim$59%) but a tractable MaxSAT instance. Even at

$T = 15$, not every random seed yields a tractable teacher, so we use a simple lottery procedure: we train many teachers with different seeds, retain those whose partition-1 MaxSAT solve completes within a 30-second timeout *and* whose test accuracy is at least 58%, and stop once 10 accepted seeds are reached. With these settings we observed an acceptance rate of approximately 4% (10 of 285 trial seeds were accepted). All matched-capacity baselines for higgs-100k are also trained at $T = 15$, $s = 3.0$, so that all methods see comparable teachers.

**Baselines.** All methods are evaluated at the same final clause count $K$ as the compressed model:

- **Matched TM (primary baseline):** Train a fresh TM with $K$ clauses, split as evenly as possible between the two classes (i.e., $\lceil K/2 \rceil$ and $\lfloor K/2 \rfloor$), from scratch on the training data. This tests whether compression from an over-parameterized model outperforms direct training at the target capacity.

- **Random pruning:** Train a 200-clause TM, then retain $K$ clauses selected uniformly at random. Predictions use clause voting as in standard TMs. This baseline tests whether MaxSAT's structured selection outperforms uninformed selection.

- **Greedy pruning:** Train a 200-clause TM, then iteratively remove the clause whose deletion causes the smallest drop in validation accuracy, until $K$ clauses remain. This tests whether global optimization (MaxSAT) outperforms local, myopic optimization.

- **Knowledge distillation:** Train a 200-clause teacher TM, generate its predictions on training samples, then train a $K$-clause student TM on these teacher labels instead of ground truth. We adapt the standard distillation framework (Hinton et al., 2015) to TMs by using hard teacher labels, since TM outputs are binary class labels rather than temperature-softened probability distributions.

**Evaluation Protocol.** We use stratified 80/20 train/test splits for all experiments. All non-higgs datasets are averaged over 10 seeds (42, 123, 456, 789, 1001, 1618, 2024, 2025, 2718, 3141). higgs-100k uses the 10 seeds returned by the lottery procedure described above. All results are reported as mean test accuracy. The compression rate is defined as $1 - K/200$, where $K$ is the median (rounded) number of clauses retained by MaxSAT. Statistical significance is assessed via the Wilcoxon signed-rank test across all 13 datasets.

**Implementation.** Our implementation uses pyTsetlinMachine 0.6.6 for TM training and python-sat 1.8.dev24 (RC2

algorithm) for MaxSAT solving. Experiments run on a Linux workstation (Ubuntu 24.04, glibc 2.39) with 96 vCPU and 756 GB RAM.

**Reproducibility.** We provide a fully pinned environment (Python 3.12.11, numpy 2.3.5, scikit-learn 1.7.2, pyTsetlinMachine 0.6.6, python-sat 1.8) together with the compiled `libTM.so` binary used in our experiments. The pyTsetlinMachine C backend uses unseeded multi-threaded `rand()`, so a freshly recompiled binary is not bit-reproducible even with the same source and seed; reusing the supplied binary on a compatible system reproduces the table numbers exactly, and on other systems the recompiled binary preserves the relative comparisons (MaxSAT vs. Matched, Random, Greedy, KD) reported here. All code, the pinned environment, the compiled binary, and the raw per-seed result JSONs are archived on Zenodo at https://doi.org/10.5281/zenodo.20217704.

### 5.2. Main Results

Table 2 compares the compressed model against the 200-clause teacher (Full TM) and against a fresh TM trained from scratch at the same clause budget (Matched).

**Compression is near-lossless.** Across all 13 datasets, MaxSAT compression preserves most of the teacher's accuracy: the gap to the Full TM is below 1pp on 11 of 13 datasets and never exceeds 4.4pp (tictactoe). On spect-heart, MaxSAT marginally exceeds the Full TM (84.4 vs 83.7), within seed-level variance. This holds despite very aggressive compression: the median retained clause count is just 16, against the teacher's 200.

**Matched-capacity direct training fails on the same problems.** Training a TM from scratch with the small clause budget that MaxSAT achieved is dramatically worse on every dataset where direct training has room to improve, with gaps up to +45pp (spect-heart) and +28pp (breast-cancer) (Figure 1). The exception is nursery, where the problem is simple enough that even a single-clause TM reaches 100% test accuracy and the two methods tie. This pattern supports the central hypothesis: small but well-chosen clause subsets exist inside an over-parameterized TM, but stochastic small-capacity training cannot find them.

**Compression rates.** Compression rates range from 83% (spambase) to over 99% (nursery), retaining between 1 and 35 clauses out of the original 200. Smaller or near-trivial datasets (banknote, nursery, spect-heart) compress most aggressively; the highest-dimensional dataset (spambase, 57 features) retains the most clauses.

*Table 2.* Test accuracy (%) over 10 seeds. **Full TM**: 200-clause teacher. **MaxSAT**: our compressed model. **Matched**: a fresh TM trained from scratch with the same $K$ clauses MaxSAT selected. $K$ is the median number of clauses MaxSAT retains; *Compr.* is the corresponding compression rate $1 - K/200$. $\Delta = $ MaxSAT $-$ Matched. Bold indicates the winner among MaxSAT and Matched.

| DATASET | FULL TM | MAXSAT | MATCHED | $\Delta$ | $K$ | COMPR. |
|---|---|---|---|---|---|---|
| SPECT-HEART | 83.7 | **84.4** | 39.1 | +45.4 | 6 | 97% |
| BREAST-CANCER | 92.2 | **91.4** | 63.5 | +27.9 | 12 | 94% |
| BANKNOTE | 86.3 | **86.3** | 71.1 | +15.2 | 4 | 98% |
| TICTACTOE | 89.1 | **84.7** | 76.9 | +7.8 | 23 | 89% |
| KR-VS-KP | 98.0 | **97.6** | 93.5 | +4.1 | 21 | 90% |
| MUSHROOM | 100.0 | **100.0** | 96.1 | +3.9 | 8 | 96% |
| CAR | 94.8 | **93.7** | 91.2 | +2.5 | 22 | 89% |
| ELECTRICITY | 71.2 | **71.2** | 69.2 | +2.0 | 14 | 93% |
| PHISHING | 78.8 | **78.8** | 77.0 | +1.8 | 17 | 92% |
| MAGIC | 78.6 | **78.5** | 77.1 | +1.4 | 16 | 92% |
| HIGGS-100K | 58.7 | **58.2** | 56.9 | +1.3 | 26 | 87% |
| SPAMBASE | 94.3 | **93.7** | 93.0 | +0.7 | 35 | 83% |
| NURSERY | 100.0 | **100.0** | 100.0 | +0.0 | 1 | >99% |

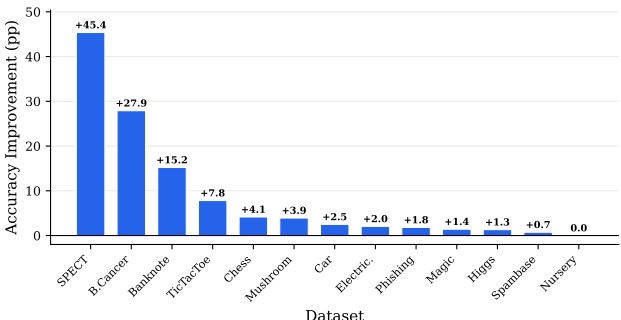

*Figure 1.* Accuracy improvement ($\Delta$) of MaxSAT compression over Matched TM baseline across 13 datasets, sorted by improvement magnitude. MaxSAT improves on or matches the matched-capacity baseline on every dataset, with gaps up to +45pp where direct small-TM training fails.

*Table 3.* Comparison of compression methods (test accuracy %). All methods start from the same 200-clause teacher and compress to the same $K$ clauses chosen by MaxSAT. Bold indicates the best result per dataset.

| DATASET | MAXSAT | RANDOM | GREEDY | KD |
|---|---|---|---|---|
| SPECT-HEART | **84.4** | 82.2 | 82.8 | 39.6 |
| BREAST-CANCER | **91.4** | 90.3 | 89.3 | 64.5 |
| BANKNOTE | **86.3** | 71.9 | 64.4 | 72.4 |
| TICTACTOE | 84.7 | **85.0** | 76.0 | 73.6 |
| KR-VS-KP | **97.6** | 94.7 | 87.0 | 94.5 |
| CAR | **93.7** | 92.6 | 91.4 | 91.0 |
| SPAMBASE | **93.7** | 92.2 | 87.3 | 92.6 |
| PHISHING | **78.8** | 75.9 | 70.6 | 76.6 |
| MAGIC | **78.5** | 77.2 | 74.9 | 77.8 |
| ELECTRICITY | 71.2 | 67.8 | 65.4 | **71.3** |
| HIGGS-100K | **58.2** | 52.6 | 52.2 | 54.2 |
| MUSHROOM | **100.0** | 95.9 | 82.8 | 94.5 |
| NURSERY | **100.0** | 74.9 | 67.8 | **100.0** |

**Statistical significance.** A Wilcoxon signed-rank test of MaxSAT vs. Matched across all 13 datasets yields $p < 0.002$.

**Scalability.** The higgs-100k result shows that the partition-and-merge strategy scales to 100,000 samples; compression completes in approximately 2 minutes. Cross-partition separation violations are negligible: on 8 of 13 datasets the per-seed median violation rate is below 0.01%, and the largest single-seed rate across all 210 runs is 2.6% on spect-heart (the smallest dataset, with only 267 samples).

### 5.3. Comparison with Pruning Methods

We compare our methods to three alternative compression approaches: random pruning, greedy pruning, and knowledge distillation. All the evaluated methods use the same 200-clause TM and compress to the same clause count $K$ determined by MaxSAT.

**Overall performance.** Table 3 and Figure 2 compare MaxSAT against three alternative post-hoc compression methods that all start from the same 200-clause teacher and reduce to the same $K$ clauses. MaxSAT is best-or-tied-best on 10 of the 12 non-higgs datasets and, on the remaining two (tictactoe and electricity), is within 0.5pp of the winning method.

**Greedy pruning underperforms across the board.** Greedy pruning is the weakest method on 9 of the 13 datasets, with gaps to MaxSAT of up to 22pp on banknote, 11pp on kr-vs-kp, and 32pp on the near-saturated nursery. As noted in Section 1, TM clauses interact non-linearly through Boolean logic and vote aggregation, so a one-clause-at-a-time procedure drops the wrong ones.

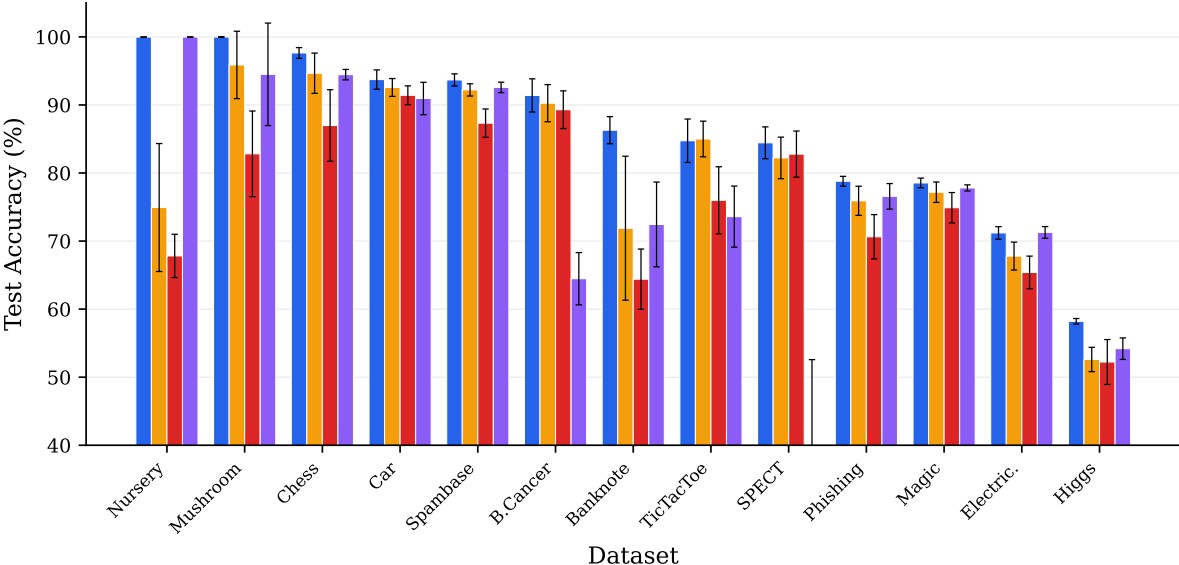

*Figure 2.* Test accuracy across compression methods. MaxSAT achieves the highest or near-highest accuracy on every dataset. Greedy pruning is the weakest method overall; knowledge distillation collapses on small datasets (spect-heart, breast-cancer) where capacity-limited stochastic training cannot succeed. **Legend:** ■ MaxSAT, ■ Random, ■ Greedy, ■ KD.

**Random pruning is a strong baseline.** Because the 200-clause teacher already contains many high-quality clauses, randomly retaining a subset of them often performs reasonably well, especially on datasets where most clauses are interchangeable (spect-heart, tictactoe). On these datasets MaxSAT's advantage over random pruning is modest. On datasets where specific clause combinations are critical (banknote, kr-vs-kp), MaxSAT's structured selection wins by a wide margin.

**Knowledge distillation fails on small datasets.** KD inherits the matched-capacity training failure mode on small datasets, with near-random accuracy on spect-heart (39.6%) and breast-cancer (64.5%). On larger datasets, KD is comparable to MaxSAT, narrowly winning on electricity (71.3 vs 71.2). This is consistent with the central hypothesis: changing the training signal alone does not overcome the small-capacity stochastic search problem.

### 5.4. Sensitivity

We assess robustness with respect to three design choices: (i) the decoder, (ii) the clause budget $K$, and (iii) the partition count $p$.

**The hybrid predictor is essential.** The compressed clauses $\mathcal{K}$ are selected to *discriminate* training pairs, not to vote correctly. As a result, predicting from $\mathcal{K}$ by clause voting alone is close to random: across the 13 datasets the $T_{\mathcal{K}}$-only test accuracy has median 50% (range 38–65%),

with a median gap of 36 pp to the hybrid predictor. The hybrid combination of $\mathcal{K}$ with the pattern dictionary $D$ recovers within a few pp of the 200-clause teacher on every dataset (see Table 2). This justifies our two-component predictor design.

**Sweeping the clause budget $K$.** The MaxSAT encoding admits an at-most-$K$ cardinality constraint, giving direct control over the compressed model size. We sweep $K \in \{5, 10, 20, 40, 80\}$ on three datasets, 10 seeds each, with the same hyperparameters as Table 2.

The pattern is consistent across datasets: MaxSAT wins decisively when $K$ is small, with the gap closing as $K$ approaches the natural minimum discriminating set size (here $\sim$10–20). Beyond that, MaxSAT saturates (adding budget cannot reduce the already-tight discriminating set), while direct training continues to benefit from added capacity. This confirms a *regime-dependent* advantage: MaxSAT compression is most useful when the target budget is tight, exactly the operating point that motivates compression.

**Sweeping the partition count $p$.** Partitioning is a scalability heuristic and we verify it does not distort the result. On the same three datasets, with $p \in \{4, 8, 16, 32, 64, 128\}$ and 10 seeds, the hybrid test accuracy is flat: standard deviation across $p$ values is below 0.5 pp on each dataset. Solve time, in contrast, drops sharply with $p$: on phishing from 7.5 s at $p = 4$ to 0.17 s at $p = 128$ (44×). Practically, $p$ controls runtime without affecting quality, so it can be set

*Table 4.* K-sweep: MaxSAT vs. Matched (test accuracy %, 10 seeds). At small budgets MaxSAT dominates; at $K = 80$ matched-capacity training catches up. "Infeas." = no $K$-clause subset can satisfy all hard constraints. [†] Averaged over the 6 feasible seeds (4 of 10 seeds infeasible at this budget).

| DATASET | $K$ | MAXSAT | MATCHED | $\Delta$ |
|---|---|---|---|---|
| BREAST-CANCER | 5 | **89.9** | 48.9 | +41.0 |
| BREAST-CANCER | 10 | **91.4** | 57.7 | +33.7 |
| BREAST-CANCER | 20 | **91.4** | 71.2 | +20.2 |
| BREAST-CANCER | 40 | **90.9** | 85.7 | +5.2 |
| BREAST-CANCER | 80 | **91.6** | 91.3 | +0.3 |
| KR-VS-KP | 5 | **95.6**[†] | 73.9 | +21.6 |
| KR-VS-KP | 10 | **96.3** | 89.9 | +6.4 |
| KR-VS-KP | 20 | **97.2** | 93.9 | +3.2 |
| KR-VS-KP | 40 | **97.2** | 95.5 | +1.7 |
| KR-VS-KP | 80 | **97.1** | 96.6 | +0.5 |
| PHISHING | 5 | INFEAS. | 57.8 | — |
| PHISHING | 10 | **78.2** | 74.1 | +4.2 |
| PHISHING | 20 | **78.9** | 77.3 | +1.7 |
| PHISHING | 40 | **79.0** | 77.9 | +1.1 |
| PHISHING | 80 | **78.9** | 78.7 | +0.2 |

as aggressively as memory allows for solver tractability.

## 6. Discussion

**Why Compression Beats Direct Training.** The experimental results are consistent with the mechanism described in Section 1: over-parameterized TMs find good solutions despite stochastic search limitations, and MaxSAT solves the resulting combinatorial optimization problem to extract minimal discriminating sets.

**The Hybrid Predictor.** The compressed model consists of the $K$ selected clauses together with a pattern dictionary of $U$ binary $K$-bit vectors and their labels, plus a 1-NN fallback for unseen patterns. Total memory is therefore proportional to $K + U \cdot K$, against the teacher's 200 clauses; with $K$ typically between 1 and 35, this corresponds to 83–99% fewer clauses, and the dictionary remains compact whenever the number of distinct training patterns $U$ is small.

**Limitations.** Our approach via the $O(n^2/p)$ constraint count appears limited in practice to datasets of roughly 100K samples; beyond this, further algorithmic improvements or approximations may be needed. Since our main objective was to establish compression for binary classification, we have not attempted to optimize multi-class settings. An extension of our approach to one-vs-one or one-vs-rest decomposition is left for future work. We have also focused on datasets with at least 300 samples; for very small datasets, the 80/20 train/validation split used for clause weighting reduces effective training data enough to increase variance.

## 7. Conclusion

This work makes three contributions:

1. We define oracle-based compression as the Minimum Discriminating Clause Set problem and prove it NP-hard.

2. We show that MDCS can be encoded as weighted partial MaxSAT with hard constraints for separation and soft constraints for minimization.

3. We observe that partition-and-merge reduces constraint complexity from $O(n^2)$ to $O(n^2/p)$, enabling compression of 100K-sample datasets in minutes.

Empirically, the compressed model attains the teacher's accuracy within 1pp on 11 of 13 datasets and never drops more than 4.4pp, while a matched-capacity TM trained from scratch collapses on small datasets (gap in MaxSAT's favor up to +45pp). The sensitivity analysis (Section 5.4) sharpens the picture: MaxSAT's advantage is concentrated at tight clause budgets near the natural minimum-discriminating-set size, and the pattern dictionary, rather than clause voting, is what turns the selected subset into a working classifier.

These results indicate that compact, high-performing clause subsets exist within over-parameterized TMs but are difficult to find through direct stochastic training at small capacity. This parallels, in the TM setting, the *lottery-ticket* intuition from neural network research, with the distinction that the Boolean structure of TM clauses makes extraction amenable to exact combinatorial optimization rather than magnitude-based heuristics.

In short: train large, then compress via MaxSAT. This separation decouples learning from selection and yields models that are smaller, nearly as accurate, and interpretable as propositional clauses.

## Acknowledgements

The research was supported by the Austrian Science Fund (FWF) projects 10.55776/COE12 (Bilateral Artificial Intelligence) and 10.55776/P36420 (Structure Identification with SAT).

## Impact Statement

Reducing Tsetlin Machine model size while preserving accuracy supports two deployment scenarios where neural networks are poorly suited: interpretable classification, in which a small set of human-readable Boolean clauses can be inspected by domain experts (e.g. for medical diagnosis or regulatory review), and inference on resource-constrained hardware such as FPGAs and microcontrollers,

where memory and energy budgets rule out large models. The compression itself is offline and post-hoc, so it inherits the safety properties of the trained teacher model and does not introduce new failure modes. Our evaluation is limited to binary classification on tabular benchmarks; extending the approach to multi-class settings, sequence data, or larger feature spaces is left to future work, and practitioners should validate any compressed model on deployment-representative data before use.

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
