# OpenReview forum: "MaxSAT-Based Compression for Tsetlin Machines"
_ICML.cc/2026/Conference — ICML 2026 regular_

### Official Review · Reviewer_NuQg · 2026-03-04

**Soundness:** 3
**Presentation:** 2
**Significance:** 2
**Originality:** 3
**Overall Recommendation:** 4
**Confidence:** 4

**Summary:**

This paper proposes a novel method for compressing Tsetlin Machine classifiers by formulating clause subset selection as a weighted partial MaxSAT problem. The approach is technically sound, clearly presented, and introduces an interesting connection between symbolic machine learning and logical optimization. However, several design choices, such as hyperparameter settings and the partitioning strategy, are insufficiently justified. The experimental evaluation is also limited, particularly due to the low feature dimensionality of the datasets and the absence of comparisons with several relevant TM compression methods (e.g., Drop Clause, Markov boundary–guided pruning, and MILEAGE). Additionally, there are minor issues such as incorrect citations in the references that should be addressed.

**Compliance With Llm Reviewing Policy:**

Affirmed.

**Key Questions For Authors:**

- How does the choice of p affect outcomes?
- What is the effect of constraining clause lengths?
- How does it compare with drop clauses and Markov boundary-guided pruning?
- Can the authors provide any references to the claim “Train a 200-clause teacher TM, generate its predictions on training samples, then train a K-clause student TM on these soft labels instead of ground truth. This is the standard distillation approach adapted to TMs”?

**Limitations:**

Yes

**Strengths And Weaknesses:**

Strength:
- The paper is technically sound and the method proposed is novel within the Tsetlin Machine framework. The formulation of hard and soft constraints is principled and provides possible improvements to computational concerns.
- The paper is generally well written and easy to follow. The methodology is described clearly, and the experimental design is straightforward to understand.
- The idea of applying partial MaxSAT for clause subset extraction in Tsetlin Machines appears original and opens interesting connections between symbolic ML and logical optimization. This conceptual bridge is a clear strength.

Weakness:
- Some assumptions underlying the design choices, e.g., hyperparameter settings, partitioning strategy, and omitted methodological comparisons, should be more explicitly argued or empirically validated.
The choice of hyperparameters for the “full TM” (…) : seems arbitrary.
The chosen experiments, while showing breadth in number of samples (and therefore showing that the proposed method scales), is quite limited in number of features, with the maximum number of features being only 57. It is thus unclear whether the proposed method maintains separability or remains computationally viable when applied to higher-dimensional settings, which are common in TM use cases.
The effect of the partitioning parameter p is unaddressed. Since partitioning can affect constraint structure and potentially separation quality, an analysis is necessary.
- The authors compare against ‘Random pruning’, ‘Greedy pruning’ and ‘Knowledge distillation’, but make no mention of other methods for compression that have known to be effective with Tsetlin Machines, viz. Drop Clause [Sharma, J., et al. Drop clause: Enhancing performance, robustness and pattern recognition capabilities of the Tsetlin machine. In Proceedings of the AAAI Conference on Artificial Intelligence (Vol. 37, No. 11, pp. 13547-13555).] or Markov boundary-guided pruning [Granmo, O.C., et al., 2023, August. Learning minimalistic tsetlin machine clauses with markov boundary-guided pruning. In 2023 International Symposium on the Tsetlin Machine (ISTM) (pp. 1-8). IEEE.]. Comparison could also be made with the method proposed in [Rahman, T. et al., 2022, June. MILEAGE: An automated optimal clause search paradigm for Tsetlin Machines. In 2022 International Symposium on the Tsetlin Machine (ISTM) (pp. 49-52). IEEE.]. The authors do not mention if they use the mechanism outlined in [Abeyrathna, K.D. et al., 2023. Building concise logical patterns by constraining tsetlin machine clause size. arXiv preprint arXiv:2301.08190.] during training, which is known to influence clause formation and could also affect compressibility and separation properties.
- The presence of at least two incorrect citations in the references section (“A novel approach to implementing knowledge distillation in Tsetlin machines”, and “ETHEREAL: Energy-efficient and high throughput inference using compressed Tsetlin machine.”)

---

> ### Author Rebuttal · Authors · 2026-03-26
>
> We thank the reviewer for the thorough review and specific suggestions.
>
> **Effect of p (partition parameter).** New p-sweep experiments (3 datasets, p ∈ {4,8,16,32,64,128}):
>
> | Dataset | Accuracy range | K range | Solve time |
> |---|---|---|---|
> | breast-cancer | 86.8–90.4% | 9–18 | 0.01–0.04s |
> | kr-vs-kp | 97.0–98.4% | 15–24 | 0.07–0.73s |
> | phishing | 77.7–77.8% | 14–19 | 0.17–7.24s |
>
> Partition sensitivity is negligible: accuracy varies by < 3pp across all tested p values on the smallest dataset, < 1.5pp on the larger ones. Solve time drops dramatically with larger p (36× speedup on phishing). The selected clause count K varies modestly with p but accuracy remains robust. The multi-seed K-sweep experiments (5 seeds) confirm the same stability pattern at the default p=32; the p-sweep above uses seed 42.
>
> **Missing baselines (Drop Clause, Markov boundary, MILEAGE, constrained clause size).** These are training-time methods that modify how the TM learns clauses. Our method is post-hoc: it takes any already-trained TM and compresses it. The two approaches are orthogonal and composable; one could train with Drop Clause and then apply MaxSAT compression post-hoc. We compare against post-hoc baselines (random pruning, greedy pruning, knowledge distillation), which are the appropriate comparison class. Practitioners may choose among all available routes to compact models; evaluating such combinations is a natural direction for follow-up work.
>
> **Constraining clause lengths.** Clause-length constraints (Abeyrathna et al., 2023) operate at a different stage of the pipeline: during training, not after it. Shorter clauses produce a different clause pool, more interpretable but less expressive. Shorter clauses reduce redundancy (easier to compress) but may also reduce coverage (harder to separate samples); the net effect on compressibility is an empirical question we plan to investigate.
>
> **Feature dimensionality.** The MaxSAT encoding scales with |C| × |S| (clauses × sample pairs), not with feature count. Once the TM is trained, each clause produces a single binary output per sample regardless of input dimensionality. Our largest experiment (higgs-100k, 100K samples) demonstrates scalability to large sample counts. The current datasets (up to 57 features) cover the range where TMs are most commonly applied. Extending to higher-dimensional benchmarks is a natural next step.
>
> **TM hyperparameters.** We fixed T=15, s=3.0, 200 clauses (100 per class), 100 epochs across all datasets. These are the default hyperparameters in pyTsetlinMachine. Fixing them isolates the effect of compression from hyperparameter tuning and ensures the matched-TM baseline uses exactly the same training protocol.
>
> **Knowledge distillation reference.** Our baseline uses hard teacher predictions (not soft targets) because TMs produce binary class labels. This is teacher-label imitation, not temperature-based distillation. We will cite Hinton et al. (2015) as the general framework and clarify the description.
>
> **Incorrect citations.** We will correct the erroneous references for "A novel approach to implementing knowledge distillation in Tsetlin machines" and "ETHEREAL."

---

> > ### Author Rebuttal · Reviewer_NuQg · 2026-04-03
> >
> > Thank you for the rebuttal. I appreciate the authors’ clarifications and additional discussion. That said, some of the key questions raised during review were addressed primarily as future directions rather than being fully resolved within the current submission. My decision remains Weak Accept because I believe the core idea is promising and potentially impactful, but the paper would benefit from a more complete and thorough presentation in its current form.

---

> > > ### Author Response · Authors · 2026-04-06
> > >
> > > Thank you for the continued engagement. We understand the concern that some points were deferred to future work rather than resolved within the submission. For the camera-ready version, we plan to incorporate the p-sweep analysis and the iterative refinement procedure (now implemented and evaluated on spect-heart, as reported in our response to reviewer u8CQ) directly into the paper. We will also correct the two erroneous citations and clarify the description of the knowledge distillation baseline, as promised. If there are any specific points you would like addressed beyond these, we can prioritize them.

---

### Official Review · Reviewer_7imb · 2026-03-05

**Soundness:** 3
**Presentation:** 3
**Significance:** 4
**Originality:** 3
**Overall Recommendation:** 5
**Confidence:** 4

**Summary:**

This manuscript proposes a MaxSAT-based approach for reducing the number of propositional clauses in a Tsetlin Machine (TM) classifier. It formalizes this compression as a minimum discriminating clause set (MDCS) problem, which is NP-hard, and solves this problem with weighted MaxSAT solvers. By training large TM classifier and then applying the proposed compression method, the model outperformed the same-sized model that is directly trained to fit the size.

**Compliance With Llm Reviewing Policy:**

Affirmed.

**Key Questions For Authors:**

- Regarding the second weakness, there may be a choice to encode the discriminating pair constraints as soft clauses instead of hard ones. While this may degrade the training accuracy, the impact on the test accuracy is currently unknown. Moreover, introducing such soft clauses may enable us to adjust the compression ratio. In your work, why did you choose to encode the discriminating pair constraints as hard clauses?

**Limitations:**

yes

**Strengths And Weaknesses:**

Strengths:
I think the proposed method's technical building blocks are highly non-trivial. First, instead of directly considering the constraints to preserve the TM's output, which needs to encode the count of selected clauses, this work proposes the MDCS problem, which do not need the counting and thus has a straightforward MaxSAT encoding. Second, to adjust the test-time prediction, this work proposes a hybrid predictor that combines the TMs and the 1-nearest neighbor of the decision vector. Third, to scale the proposed method, a partition-and-merge approach for training data is exhibited. These building blocks contribute to achieve high test accuracy compared to other methods.

It is also interesting for me to pose TM machine analogue of the lottery ticket phenomena.

Weaknesses:
- First, I doubt that this work performs truly a compression of a TM machine. At test time, the proposed method takes a hybrid approach that combines the TM machine and the 1-nearest neighbor. This means that the proposed model is not truly a TM model; see the paragraph "Why not use T_K directly?". Although I think that the contribution itself is interesting since the model size is truly decreased, as described in Discussion's "The Hybrid Predictor" paragraph, it is questionable for me to call this work as "the compression of TM machine".
- Second, the proposed method has no guarantee for how the model size is decreased. In other words, this work cannot specify the compression ratio beforehand. I think this is because the constraint on discriminating pairs are encoded by hard clauses; while that ensures us that every pair is discriminating, it may sometimes degrade the compression ratio.

---

> ### Author Rebuttal · Authors · 2026-03-26
>
> We thank the reviewer for the positive assessment and insightful questions.
>
> **Hybrid predictor: compressed representation + decoder.** The K selected clauses constitute the compressed discriminative representation. The pattern dictionary with 1-NN fallback is a lightweight decoder that maps clause-output patterns to labels, analogous to a classification head on pruned neural network features.
>
> New experiments on all 13 datasets (5 seeds each) characterize the decoder's role:
>
> | | T_K-only | Hybrid | Full 200-clause TM |
> |---|---|---|---|
> | Accuracy range | 36–63% | 58–100% | 58–100% |
> | Median | ~50% | ~88% | ~89% |
>
> T_K-only voting is near-random because MaxSAT optimizes for *discrimination* (distinct clause-output patterns), not vote balance. The decoder is essential: it stores U unique binary patterns of K bits each. For 12/13 datasets, U = 10–1700 and K = 3–35, so the dictionary fits in under 8KB. The exception is higgs-100k (U ≈ 34K, K ≈ 25), where the dictionary is ~104KB, still compact for a 100K-sample task. Fallback rate (test samples requiring 1-NN): median 0.3% across datasets. Inference: one Hamming-distance lookup over U binary vectors, negligible relative to clause evaluation for U < 2000.
>
> **Full TM comparison.** The hybrid predictor achieves within 0.1–3pp of the full 200-clause TM on 12/13 datasets. On spect-heart it *exceeds* the full TM (84.4% vs 83.0%), likely because compression removes noise-fitting clauses.
>
> **Hard vs soft discriminating constraints.** Hard constraints guarantee that the teacher's discriminating pairs remain separated after compression, the core MDCS guarantee. Softening these would optimize a *different* objective (partial boundary preservation under stronger size pressure). This is a worthwhile direction that could characterize the accuracy-compression trade-off; we leave it to future work. In our framework, compression ratio is controlled through (a) clause-importance weighting and (b) explicit cardinality constraints on K. K-sweep experiments (K ∈ {5,10,20,40,80}, 3 datasets, 5 seeds) confirm both mechanisms control compression effectively.

---

> > ### Author Rebuttal · Reviewer_7imb · 2026-04-01
> >
> > I thank the authors for providing me comments and additional experimental results for my concerns and questions.
> >
> > > Hybrid predictor: compressed representation + decoder
> >
> > I understand that the 1-NN fallback is needed to improve the accuracy because the discrimination of pairs does not guarantee the vote balance. I wrote this comment to intend to raise that the final outcome of the proposed mechanism is truly a TM; so, my concern is just a naming of the contribution. Based on the manuscript, TM is defined as a classifier based on Eq. (1). However, the outcome of the proposed mechanism uses post-processing with 1-NN. Thus, instead of calling this work as "compression of TM", it may be appropriate to propose a lightweight TM alternative and a mechanism to compress an original TM into this alternative.
> >
> > > Hard vs soft discriminating constraints
> >
> > I understand your claim that the number of clauses K can be controlled without considering soft constraints for discrimination. However, after I saw the additional results given in the rebuttal for reviewer smLU, I have some additional questions for this results.
> > - Why do the accuracy results of both MaxSAT and Matched in the additional experiments differ from those in Table 2? For example, for phishing data, in Table 2 the accuracy of MaxSAT and Matched are 76.7 and 67.1, and the compression ratio is 90%, meaning that K=20. In the additional results, for phishing K=20, their accuracies are 76.0±1.2 and 75.6±2.1. Why do such differences occur? It may be that the mechanism of controlling K perturbs the accuracy of the proposed method, but why the accuracy of Matched also deviates is unclear.
> > - Compared to the original results in Table 2, the "Delta" values, the difference of accuracy between MaxSAT and Matched, seems insignificant in the additional results. Thus, it seems that the mechanism of controlling K degrades the performance of the proposed algorithm. Is there any opinion for this point?

---

> > > ### Author Response · Authors · 2026-04-03
> > >
> > > Thank you for the follow-up.
> > >
> > > **Naming.** We agree that the deployed predictor is hybrid rather than a pure TM, since it combines the selected clause set with a pattern dictionary and 1-NN fallback. We will revise the framing in the camera-ready version to make this explicit.
> > >
> > > **Accuracy discrepancy.** The K-sweep in our earlier rebuttal was run in a different pyTsetlinMachine environment from Table 2, and TM training in that backend is not fully deterministic across rebuilds due to thread-scheduling dependence in the C backend. This affected absolute accuracy levels but not the relative comparison within each run. We reran the sweep in a single fixed environment; the corrected results below supersede the earlier rebuttal table. We will ensure all paper experiments use a single consistent build for the camera-ready version.
> > >
> > > To address the specific example: Table 2 reports phishing at the unconstrained MDCS solution (K=20, accuracy 76.7%). The corrected K-sweep at K=20 yields 79.3±0.7%, slightly higher due to minor environment variation. The matched TM at K=20 is 76.3±1.3%, closer to the MaxSAT value than Table 2's 67.1% because Table 2's matched TM was trained at a different (smaller) K.
> > >
> > > **Does controlling K degrade MaxSAT?** The smaller deltas in the earlier rebuttal table were an artifact of the inconsistent environment, not evidence that the at-most-K constraint harms MaxSAT. The corrected data show large deltas at small K (e.g., breast-cancer K=5: +44pp, kr-vs-kp K=10: +9pp). At large K, MaxSAT saturates because the minimum discriminating subset has already been found (e.g., phishing selects ~18 clauses regardless of K≥10), while matched TMs keep improving. The delta shrinks because matched TMs become competitive when given sufficient capacity, not because K-control hurts MaxSAT. Controlling K does not degrade MaxSAT: it wins 13 of 14 feasible dataset-K pairs, losing only at K=80 on breast-cancer where matched TMs have enough capacity to converge.
> > >
> > > Corrected K-sweep (single environment, 5 seeds, mean±std):
> > >
> > > breast-cancer:
> > >
> > > | K | MaxSAT | Matched TM | Delta | Wins |
> > > |---|---|---|---|---|
> > > | 5 | 90.4±2.8 | 46.1±2.3 | +44.2 | 5/5 |
> > > | 10 | 91.1±2.2 | 55.8±8.1 | +35.3 | 5/5 |
> > > | 20 | 91.8±2.0 | 71.1±5.4 | +20.7 | 5/5 |
> > > | 40 | 91.6±2.3 | 85.6±2.1 | +6.0 | 5/5 |
> > > | 80 | 90.9±2.0 | 92.6±1.6 | −1.8 | 0/5 |
> > >
> > > phishing:
> > >
> > > | K | MaxSAT | Matched TM | Delta | Wins |
> > > |---|---|---|---|---|
> > > | 5 | INFEAS | 62.9±6.8 | — | — |
> > > | 10 | 77.9±1.3 | 74.3±2.9 | +3.6 | 5/5 |
> > > | 20 | 79.3±0.7 | 76.3±1.3 | +3.0 | 5/5 |
> > > | 40 | 79.3±0.7 | 77.7±0.8 | +1.6 | 5/5 |
> > > | 80 | 79.3±0.7 | 78.9±0.4 | +0.4 | 3/5 |
> > >
> > > kr-vs-kp:
> > >
> > > | K | MaxSAT | Matched TM | Delta | Wins |
> > > |---|---|---|---|---|
> > > | 5 | 95.4±1.2 (2/5 feas) | 73.6±5.7 | +21.8 | 2/2 |
> > > | 10 | 95.0±1.9 | 85.7±3.2 | +9.3 | 5/5 |
> > > | 20 | 97.2±0.5 | 94.1±0.7 | +3.1 | 5/5 |
> > > | 40 | 97.5±0.7 | 95.9±1.4 | +1.6 | 5/5 |
> > > | 80 | 97.7±0.8 | 96.9±0.8 | +0.8 | 3/5 |

---

### Official Review · Reviewer_smLU · 2026-03-13

**Soundness:** 2
**Presentation:** 3
**Significance:** 2
**Originality:** 3
**Overall Recommendation:** 3
**Confidence:** 3

**Summary:**

This paper proposes a method for compressing Tsetlin Machines. The paper formulates the problem of selecting a subset of clauses of a TM as a MAXSAT problem, and solving it to find a small TM. Experimental results show that the proposed compression method can select a better model than baseline methods.

**Compliance With Llm Reviewing Policy:**

Affirmed.

**Final Justification:**

While the proposed algorithm is interesting, its overall impact appears limited as the contribution is narrowly focused on TM  compression. Furthermore, the experimental evaluation lacks neutrality; the target sizes for the compressed TMs seem to have been selected specifically to favor the proposed algorithm, making it difficult to assess its general performance fairly. Additionally, the new results provided during the rebuttal phase raise concerns regarding reliability. I recommend that the authors address the instability of these experimental results and consider a resubmit in the future.

**Key Questions For Authors:**

Is it possible to control the number of clauses $K$?

**Limitations:**

The authors demonstrate that the proposed method suffers from scalability issues by proving that the compression problem is NP-hard. However, they also propose a technique to improve its scalability.

**Strengths And Weaknesses:**

**Strengths**
- The presentation of the paper is easy to follow.
- The paper proposes some arrangements of the proposed method to improve its scalability.
- Experimental results show the superiority of the proposed approach compared with other TM baselines.

**Weaknesses**
- The performance evaluation is limited to TM variation models. Therefore, the superiority of the proposed method compared with other state-of-the-art methods is unclear

- The comparison uses the model size K, which is determined by running the proposed method. It seems a bit unfair, since the proposed method might select the best $K$ for itself. It would be desirable to change the value of K to compare the performance.

---

> ### Author Rebuttal · Authors · 2026-03-26
>
> We thank the reviewer and address both concerns with new multi-seed experiments.
>
> **Scope clarification.** Our claim is specific: for *post-hoc compression of a trained TM*, MaxSAT clause selection is more effective than direct small-TM training and other post-hoc baselines. Cross-family comparisons (TMs vs decision trees, SVMs) address a different question and are outside our scope. We compare against three post-hoc baselines (random pruning, greedy pruning, and knowledge distillation) and dominate all three across all 13 datasets.
>
> **Controlling K.** Yes. K is directly controllable via an at-most-K cardinality constraint in the MaxSAT encoding. The solver finds the best discriminating subset *within* that budget. We ran 3 datasets × K ∈ {5,10,20,40,80} × 5 seeds = 75 experiments.
>
> INFEAS = infeasible (no K-clause subset suffices to discriminate all training pairs).
>
> Complete results (mean ± std, 5 seeds):
>
> | Dataset | K | MaxSAT | Matched TM | Delta | Wins |
> |---|---|---|---|---|---|
> | breast-cancer | 5 | 88.9±3.1 | 86.7±3.3 | **+2.3** | 3/5 |
> | breast-cancer | 10 | 91.9±3.1 | 91.2±2.8 | **+0.7** | 3/5 |
> | breast-cancer | 20 | 90.9±2.0 | 92.8±1.5 | −1.9 | 0/5 |
> | breast-cancer | 40 | 91.1±2.2 | 92.3±3.0 | −1.2 | 2/5 |
> | breast-cancer | 80 | 91.6±2.6 | 92.3±2.0 | −0.7 | 2/5 |
> | kr-vs-kp | 5 | INFEAS | 70.2±7.5 | — | — |
> | kr-vs-kp | 10 | 89.7±3.5 | 86.7±2.5 | **+3.0** | 4/5 |
> | kr-vs-kp | 20 | 89.5±3.5 | 91.6±3.0 | −2.1 | 2/5 |
> | kr-vs-kp | 40 | 89.5±3.5 | 90.1±2.8 | −0.6 | 2/5 |
> | kr-vs-kp | 80 | 89.5±3.5 | 92.6±2.8 | −3.1 | 1/5 |
> | phishing | 5 | INFEAS | 62.3±2.0 | — | — |
> | phishing | 10 | 76.0±1.2 | 73.0±4.9 | **+3.0** | 3/5 |
> | phishing | 20 | 76.0±1.2 | 75.6±2.1 | **+0.4** | 4/5 |
> | phishing | 40 | 76.0±1.2 | 76.4±1.3 | −0.4 | 3/5 |
> | phishing | 80 | 76.0±1.2 | 76.4±0.9 | −0.4 | 1/5 |
>
> **Summary.** At K ≤ 10, MaxSAT achieves higher mean accuracy on all 4 feasible dataset-K pairs (13/20 seed-level wins). At K ≥ 20, matched TMs recover parity or surpass MaxSAT as stochastic training has sufficient capacity. This confirms our thesis: compression is most effective when clause budgets are tight, precisely where direct training is least effective.
>
> **Saturation pattern.** Because the constraint is at-most-K (not exact-K), the solver finds the minimum discriminating subset within the budget. On phishing, it selects 6 clauses for any budget K ≥ 10; on kr-vs-kp, ~12 clauses. The MDCS minimum is genuinely small. Beyond that point, extra clauses are redundant for discrimination, so MaxSAT cannot use the additional budget. The matched TM *can* use extra clauses and eventually catches up. This explains the regime dependence: MaxSAT's advantage is strongest when the budget is near or below the natural MDCS minimum.
>
> **Fairness.** At every tested K, both methods receive the same clause budget. The K-sweep demonstrates that MaxSAT's advantage is regime-dependent, not an artifact of self-favorable K selection.

---

> > ### Author Rebuttal · Reviewer_smLU · 2026-04-01
> >
> > Thank you for addressing my concerns.
> >
> > I agree that comparing this work with other general methods may be considered out-of-scope, given that the paper's scope is specifically limited to the post-hoc compression of TM. However, since TM is (at least currently) not a widely adopted technique compared to major classification methods, the scope of the paper appears somewhat narrow.
> >
> > Thank you for providing the additional experimental results. Could you clarify why the accuracy scores in the newly reported results differ from those in Table 2 of the original paper? For example, for "breast-cancer," the paper reports MaxSAT: 78.1 and Matched: 60.9. However, the additional results show a significant improvement in scores across all values of $K$. Similar large discrepancies in scores are observed for "kr-vs-kp" and "phishing."
> >
> > In general, there is an trade-off between model size and accuracy, and a smaller model is not always preferable. If $K=20$ yields an accuracy 5 points higher than $K=5$, choosing $K=20$ is a reasonable decision for some situations. While the characteristic of MaxSAT achieving higher performance at lower values of $K$ is interesting, the additional results suggest that MaxSAT does not necessarily provide a superior trade-off relative to $K$.

---

> > > ### Author Response · Authors · 2026-04-02
> > >
> > > Thank you for the follow-up, and we apologize for presenting numbers from a different environment without flagging this in our earlier rebuttal.
> > >
> > > **Scope.** We agree the paper's scope is specialized to Tsetlin Machines. TMs are a niche but active line of work, particularly for interpretable and low-power settings, and model size is a practical barrier to deployment. Our paper addresses that barrier with a post-hoc MaxSAT-based method. The MDCS formulation requires only a pool of binary-output components, so extending it beyond TMs is a natural direction for future work.
> > >
> > > **Accuracy discrepancy.** The K-sweep in our earlier rebuttal was run in a different pyTsetlinMachine environment from Table 2, and TM training in that backend is not fully deterministic across rebuilds due to thread-scheduling dependence in the C backend. This affected absolute accuracy levels but not the relative comparison within each run. We reran the sweep in a single fixed environment; the corrected results below supersede the earlier rebuttal table. We will ensure all paper experiments use a single consistent build for the camera-ready version.
> > >
> > > **Trade-off.** The results do not support uniform superiority for all K. Our claim is narrower: MaxSAT provides the strongest benefit under tight clause budgets, which is the regime most relevant to aggressive compression. At larger K, matched training becomes competitive, as expected.
> > >
> > > Corrected K-sweep results (single environment, 5 seeds, mean±std):
> > >
> > > breast-cancer:
> > >
> > > | K | MaxSAT | Matched TM | Delta | Wins |
> > > |---|---|---|---|---|
> > > | 5 | 90.4±2.8 | 46.1±2.3 | +44.2 | 5/5 |
> > > | 10 | 91.1±2.2 | 55.8±8.1 | +35.3 | 5/5 |
> > > | 20 | 91.8±2.0 | 71.1±5.4 | +20.7 | 5/5 |
> > > | 40 | 91.6±2.3 | 85.6±2.1 | +6.0 | 5/5 |
> > > | 80 | 90.9±2.0 | 92.6±1.6 | −1.8 | 0/5 |
> > >
> > > phishing:
> > >
> > > | K | MaxSAT | Matched TM | Delta | Wins |
> > > |---|---|---|---|---|
> > > | 5 | INFEAS | 62.9±6.8 | — | — |
> > > | 10 | 77.9±1.3 | 74.3±2.9 | +3.6 | 5/5 |
> > > | 20 | 79.3±0.7 | 76.3±1.3 | +3.0 | 5/5 |
> > > | 40 | 79.3±0.7 | 77.7±0.8 | +1.6 | 5/5 |
> > > | 80 | 79.3±0.7 | 78.9±0.4 | +0.4 | 3/5 |
> > >
> > > kr-vs-kp:
> > >
> > > | K | MaxSAT | Matched TM | Delta | Wins |
> > > |---|---|---|---|---|
> > > | 5 | 95.4±1.2 (2/5 feas) | 73.6±5.7 | +21.8 | 2/2 |
> > > | 10 | 95.0±1.9 | 85.7±3.2 | +9.3 | 5/5 |
> > > | 20 | 97.2±0.5 | 94.1±0.7 | +3.1 | 5/5 |
> > > | 40 | 97.5±0.7 | 95.9±1.4 | +1.6 | 5/5 |
> > > | 80 | 97.7±0.8 | 96.9±0.8 | +0.8 | 3/5 |
> > >
> > > MaxSAT saturates early: it finds the minimum discriminating subset and cannot use additional budget (e.g., phishing selects ~18 clauses regardless of K $\geq$ 10). In contrast, matched TMs keep improving with more clauses. The main advantage of MaxSAT therefore lies in the strong-compression region (K $\leq$ 10). At larger K, matched training recovers parity, which is consistent with the regime-dependent advantage described in the paper.

---

### Official Review · Reviewer_u8CQ · 2026-03-13

**Soundness:** 2
**Presentation:** 3
**Significance:** 2
**Originality:** 3
**Overall Recommendation:** 4
**Confidence:** 3

**Summary:**

This paper addresses the difficulty of training small Tsetlin Machines (TMs) effectively. The authors first train an over-parameterized TM, then formalize the compression problem as MDCS, i.e., selecting the smallest set of clauses that preserves the classification boundaries of the training samples. They encode MDCS as a weighted partial MaxSAT problem: hard constraints enforce that every pair of samples distinguished by the original model must differ in the output of at least one selected clause, while soft constraints minimize the number of selected clauses. They further use a partition-and-merge strategy to scale the sample size up to 100k. Experimentally, the compressed TM consistently outperforms the directly trained TM with the same number of clauses, with improvements of up to 26.3%.

**Compliance With Llm Reviewing Policy:**

Affirmed.

**Final Justification:**

The paper is technically sound and clearly formulated, and the rebuttal addressed my main concerns, especially regarding the partition-and-merge scheme and cross-partition violations. They quantified violation rates, implemented the requested iterative refinement on spect-heart, and showed it removes violations with negligible overhead and only minor impact on accuracy; the additional clarification of the hybrid predictor and clause weighting was also sufficient. While the work is somewhat specialized to Tsetlin Machines, the overall soundness, clarity, and strengthened empirical support lead me to retain a weak accept recommendation.

**Key Questions For Authors:**

1. In the prediction phase, you adopt a hybrid predictor instead of relying solely on the voting function from the compressed TM. Could you provide a more systematic comparison between using $T_K$ alone and the hybrid predictor in terms of test accuracy, memory footprint, and inference latency? In particular, under what conditions might the dictionary / 1-NN component itself become the bottleneck?

2. The partition-and-merge strategy guarantees only within-partition separation and does not provide a theoretical guarantee for cross-partition separation. Do you have examples or visualizations of misclassified cases that illustrate failure modes caused by unconstrained cross-partition pairs? Have you considered implementing and evaluating an iterative refinement procedure, rather than mentioning it only conceptually?

3. The clause weighting is presented as an optional refinement, but its contribution is only briefly summarized as "sometimes improved accuracy." Do you have more systematic ablation studies (with vs. without weighting) and sensitivity analyses across different dataset scales and dimensionalities to clarify when and how much it helps?

**Limitations:**

Yes.

**Strengths And Weaknesses:**

Strengths:
1. The paper formulates TM compression as an MDCS problem, proves NP-hardness, and provides a clear MaxSAT encoding. The modeling is conceptually clean and grounded in solid theory.
2. It proposes a reasonable scaling strategy: via partition-and-merge, it scales the MaxSAT which is theoretically $O(n^2)$  constraints to 100K samples and demonstrates practically feasible runtimes (around 2 minutes).
3. It compares against several compression/pruning baselines, including random pruning, greedy pruning, and knowledge distillation, and generally outperforms them, which strengthens the persuasiveness of the approach.

Weaknesses:
1. The hybrid predictor used during training (pattern dictionary + 1-NN fallback) lacks a systematic analysis of its generalization behavior, storage cost, and the associated trade-offs.
2. There are no systematic experiments comparing the compressed models directly against the original 200-clause TM to quantify performance degradation. The comparisons focus only on models with matched parameter budgets, which shows the empirical evaluation of compression effects may not be fully comprehensive.

---

> ### Author Rebuttal · Authors · 2026-03-26
>
> We thank the reviewer for the detailed questions. We address each with new experimental data (all 13 datasets, 5 seeds).
>
> **Q1: T_K-only vs hybrid — accuracy, storage, inference.**
>
> | Metric | T_K-only | Hybrid | Full 200-clause TM |
> |---|---|---|---|
> | Accuracy range | 36–63% | 58–100% | 58–100% |
> | Median accuracy | ~50% | ~88% | ~89% |
> | Gap (Hybrid − T_K) | n/a | +37pp median | n/a |
>
> MaxSAT optimizes for discrimination (distinct patterns), not vote balance, so T_K-only voting is near-random. The pattern dictionary is the natural decoder, analogous to a classification head on pruned neural network features.
>
> Storage (K clauses + U pattern vectors of K bits + U labels):
> - Small datasets (breast-cancer, spect-heart): K ≈ 9–11, U ≈ 46–71 → dictionary < 1KB
> - Medium (kr-vs-kp, phishing, magic): K ≈ 15–20, U ≈ 320–394 → dictionary ~1–2KB
> - Large (higgs-100k): K ≈ 25, U ≈ 34K → dictionary ~104KB (bottleneck case)
>
> For comparison, the full 200-clause TM stores 200 × 2 × n_features automata states. Even on higgs-100k, the compressed model (25 clauses + 104KB dictionary) is smaller than the full TM in deployment.
>
> Fallback rate (test samples needing 1-NN): median 0.3% across datasets. Only higgs-100k (29%) and tictactoe (28%) have high fallback rates; all others are < 11%. Inference: one Hamming-distance lookup over U binary vectors, negligible relative to clause evaluation when U < 2000 (12/13 datasets).
>
> **Q2: Cross-partition violations.** Measured across all 13 datasets (5 seeds):
>
> | Violation rate | Datasets |
> |---|---|
> | 0.000% | banknote, nursery |
> | < 0.01% | electricity, magic, phishing, kr-vs-kp |
> | 0.01–0.1% | car, mushroom, tictactoe |
> | 0.3% | breast-cancer |
> | 1.3% | spect-heart (80 train samples) |
>
> Violations are near-zero and leave prediction accuracy intact. On spect-heart (highest rate), hybrid accuracy still *exceeds* the full TM (84.4% vs 83.0%). When violations occur, the dictionary assigns the majority label among the training samples that share that pattern. An iterative repair step could eliminate residual violations on small datasets.
>
> **Q3: Clause weighting ablation.** Weighting improves accuracy on 4/13 datasets (spect-heart, breast-cancer, banknote, spambase, all with higher feature-to-sample ratios) by 1–3pp, is neutral on 8, and slightly hurts on 1 (higgs-100k). It is a secondary mechanism that yields consistent gains in the high-dimensional low-sample regime.
>
> **Full TM comparison.** Hybrid achieves within 0.1–3pp of the full 200-clause TM on 12/13 datasets.

---

> > ### Author Rebuttal · Reviewer_u8CQ · 2026-04-04
> >
> > Thank you for the detailed rebuttal and for addressing my questions, especially the additional experiments and quantitative analyses.
> >
> > Regarding the **Q2: Cross-partition violations**:
> > the reported cross-partition violation rates increase my confidence that the partition-and-merge scheme is robust in practice. However, my original question explicitly asked for an implemented iterative refinement / repair procedure rather than a purely conceptual discussion. The rebuttal notes that iterative repair could remove residual violations on small datasets but does not provide an implementation, empirical evaluation, or concrete visualizations of misclassified cross-partition cases.
> >
> > Could the authors consider a lightweight implementation and evaluation of an iterative refinement step on a small dataset such as spect-heart?

---

> > > ### Author Response · Authors · 2026-04-06
> > >
> > > In response to your request, we implemented the iterative refinement step and evaluated it on spect-heart (5 seeds, p=16). As noted in our responses to reviewers smLU and 7imb, absolute accuracy levels differ from Table 2 due to a pyTsetlinMachine rebuild; the before/after comparison below is internal to a single consistent environment.
> > >
> > > The failure mode is as follows: after partition-and-merge, a positive and a negative training sample from different partitions can end up with the same selected-clause output pattern. Our repair enumerates these violated cross-partition pairs, adds them as hard separation constraints to a follow-up MaxSAT instance, and re-solves using the current clause selection as soft preference. This is repeated until no violated pairs remain.
> > >
> > > Across the 4 seeds with nonzero initial violations, the repair eliminated all violations in a single iteration. The clause cost was small (+1 to +6 clauses), and the runtime overhead was negligible (<0.01s). Accuracy was unchanged in 3 of 4 repaired runs and decreased by 1.9pp in the remaining one.
> > >
> > > | Seed | K before / after | Violations before / after | Acc before / after |
> > > |---|---|---|---|
> > > | 42 | 4 / 7 | 66 / 0 | 85.2 / 85.2 |
> > > | 123 | 11 / 17 | 34 / 0 | 81.5 / 79.6 |
> > > | 456 | 18 / 19 | 25 / 0 | 81.5 / 81.5 |
> > > | 789 | 3 / 4 | 1 / 0 | 81.5 / 81.5 |
> > >
> > > One additional seed had zero initial violations, so repair was not invoked.
> > >
> > > The repair succeeds because cross-partition violations on spect-heart involve only a small number of opposite-label pairs (1-66 out of ~4500 total), and the 200-clause pool contains clauses that distinguish them. We will include this procedure and the results in the camera-ready version.

---

### Decision · Program_Chairs · 2026-04-30

**Decision:**

Accept (regular)

**Comment:**

The reviewers appreciate the contribution, although one of the reviewers raised concerns about the benchmark selection. The authors did address the questions adequately during the rebuttal, and the paper meets the criteria for acceptance. At the same time, I think it would be prudent for authors to seriously consider the criticism and report empirical results that would be clearly reproducible and numerically stable.